# Effects of joint mobilization combined with acupuncture on pain, physical function, and depression in stroke patients with chronic neuropathic pain: A randomized controlled trial

Ji-Eun Lee[1], Takayuki Akimoto[2], Jisuk Chang[3], Ho-Seong Lee[1,4]*

1 Department of Exercise and Medical Science, Dankook University, Cheonan, Republic of Korea,
2 Laboratory of Muscle Biology, Faculty of Sport Sciences, Waseda University, Tokorozawa, Japan,
3 Department of Sports Management, Dankook University, Cheonan, Republic of Korea, 4 Institute of Medical-Sports, Dankook University, Cheonan, Republic of Korea

* hoseh28@dankook.ac.kr

## Abstract

### Objective

To investigate the effectiveness of joint mobilization (JM) combined with acupuncture (AC) for the treatment of pain, physical function and depression in poststroke patients.

### Methods

A total of 69 poststroke patients were randomly assigned to the JM+AC group (n = 23), the JM group (n = 23), and the control group (n = 23). Patients in the JM+AC group and the JM group received JM for 30 minutes, twice a week for 12 weeks, and the JM+AC group received AC for 30 minutes separately once a week. The control group did not receive JM or AC. Pain (visual analog scale, shoulder pain and disability index, Western Ontario and McMaster universities osteoarthritis index), physical function (range of motion, 10-m walking speed test, functional gait assessment, manual function test, activities of daily living scale, instrumental activities of daily living scale), and depression (center for epidemiologic studies depression scale, Beck depression inventory) were assessed for each patient before and after the 12 weeks of intervention.

### Results

Pain and physical function were improved significantly in the JM+AC group compared with the JM and control groups. Physical function and depression were improved significantly in the JM+AC and JM groups compared with the control group.

**Data Availability Statement:** All data files are available from the public repository Figshare

database (https://doi.org/10.6084/m9.figshare.21789092).

**Funding:** The authors received no specific funding for this work.

**Competing interests:** The authors have declared that no competing interests exist.

## Conclusion

The treatment of JM combined with AC improved pain, depression, and physical function of poststroke patients with chronic neuropathic pain in this study. This valuable finding provides empirical evidence for the designing therapeutic interventions and identifying potential therapeutic targets.

## Introduction

Stroke is one of the most serious medical issues in the world, leaving more than 80 million survivors disabled [1]. Physical disability is a common symptom in patients with chronic stroke lasting longer than 6 months, and more than half of these patients have trouble performing activities of daily living (ADL) due to severe neurological impairment [2]. Only a small percentage of stroke victims can regain hand function in the early stages of the illness, and poststroke patients need assistance to participate in independent activities in the community due to the severity of their neurological impairment [3]. Due to their diminished cognitive function and impaired gait, post-stroke patients are more likely to experience depression and have a lower quality of life [4]. Furthermore, within 2 years of a stroke, poststroke patients develop chronic pain due to nerve damage, in addition to physical function and daily living performance, which develops into muscle stiffness, spasms, and hormonal changes, which are major challenges for poststroke patients recovery [5, 6].

Shoulder pain experienced by more than 50% of poststroke patients, is a common sequelae [6], interfering with rehabilitation and reducing arm function and ADL [7]. Knee pain is the second most common pain in patients after stroke, and it has been reported to cause motor dysfunction and abnormal gait [8]. Given that most poststroke patients have decreased gait velocity and that independent gait is essential for performing ADL after a stroke, the restoration of gait function is a key objective of rehabilitation [9]. Additionally, sensorimotor dysfunction causes limited joint range of motion (ROM) and muscle weakness in the lower extremities of the affected side, leading to difficulties in performing functional activities, such as sit to stand and gait [10, 11]. Therefore, it is necessary to reduce pain and increase the joint ROM to improve physical function in poststroke patients. To tackle these problems and to help restore and improve the function of upper and lower limbs, a wide range of treatments has been proposed to therapists, so far.

Joint mobilization (JM) is an orthopaedic manipulation technique that uses the convex-concave rule and third stage of translatoric movement to alleviate contractures, enhance mobility, and reduce pain [12, 13]. The convex-concave rule states that when gliding on a convex joint surface, the direction of movement should be opposite to that of the bone, whereas when gliding on a concave joint surface, the direction of movement should be in the same direction as the bone. When a convex joint surface is distracted, the joint must be separated; in contrast, when a concave joint surface is distracted, the joint is pulled along the long axis of the bone. [12, 13]. JM works mechanically to break up the contracture by moving the joint directly, and it works arthrokinetically to suppress or help the related muscles by stimulating the joint receptors [14]. However, there appears to be a limit to how JM alone can reduce pain and restore physical function in poststroke patients [15].

Acupuncture (AC) is commonly used to treat/manage chronic diseases in Korea [16]. It is recognized to be effective in regulating physiological balance by controlling muscles and acupuncture points or by stimulating the affected area [17]. And AC is effective in reducing pain

by stimulating the nerves in stroke patients and help to restore motor function and improve the ADL score [18]. Studies suggest that AC is also effective in improving pain and depression in poststroke patients [19]. Particularly, AC combined with other therapies is more effective in poststroke patients than AC alone [20]. Therefore, it is possible that AC combined with JM is more effective to improve pain, depression, and physical function in poststroke patients. Although there are many studies about effectiveness of JM for poststroke patients, a few studies have examined the effects of JM combined with AC for these patients.

In this work, we examined the changes in pain, physical function, and depression before and after the 12 weeks of intervention of JM combined with AC in poststroke patients. We hypothesized that combining JM with AC would improve physical function more compared with JM alone.

## Methods

### 1. Study design

The study had a random sampling design (Fig 1). Participants were randomized electronically using block sizes of 3 across groups. All investigators and data analysts were blinded until the study and analysis were completed.

For the two-way ANOVA analysis involving three groups, we determined the appropriate sample size using G*Power software (version 3.1.9.7) [21]. The calculation incorporated an effect size of 0.25, an alpha level of 0.05, a power of 0.80, three groups, two measures, and a correlation of 0.5 among repeated measures. Based on the analysis, a recommended sample size of 42 participants was determined. Considering an anticipated dropout rate of approximately 50%, a total of 69 participants were recruited for the study.

Written informed consent was obtained from all subjects and the signed informed consent documents were kept on file. The protocol was approved by the Dankook University ethics committee (IRB number: DKU-2021-05-046) and it adhered to the tenets of the Declaration of Helsinki and its later amendments or comparable ethical standards. The individual in this manuscript has given written informed consent (as outlined in PLOS consent form) to publish these case details.

The recruitment period for this study was from June 10 to July 14, 2021, and the follow-up period was from June 14, 2021 to September 10, 2021. No dropouts or adverse events were reported. Although our institutions ethics committee did not require clinical trial registration, the study was retrospectively registered. We attempted to register the study as a clinical trial in October 2022, and we were successfully retroactively registered in the registry on November 22, 2022. This study conformed the protocol exactly until the study completion, with no deviations in methodology and only minor changes to the research title.

This clinical trial was registered with the Clinical Research Information Service (CRIS) of Republic of Korea approved by the WHO. Registration number: KCT0007924. Date registered: November 22, 2022, retrospectively registered. The authors confirm that all ongoing and related trials for this drug/intervention are registered.

### 2. Participants

In this study, 69 adults with stroke diagnoses who were enrolled in a public health and welfare center in South Korea's D city were included. This group was further divided into the JM+AC, the JM, and the control groups (n = 23 each). The inclusion criteria were: diagnosis of stroke ≥ 6 months ago; pain in the shoulders and knees for more than 6 months; and a Korean mini mental state examination score of ≥ 24. The exclusion criteria were: a risk of tumors or infections; a previous history of orthopedic disease or ankle surgery; a previous history of

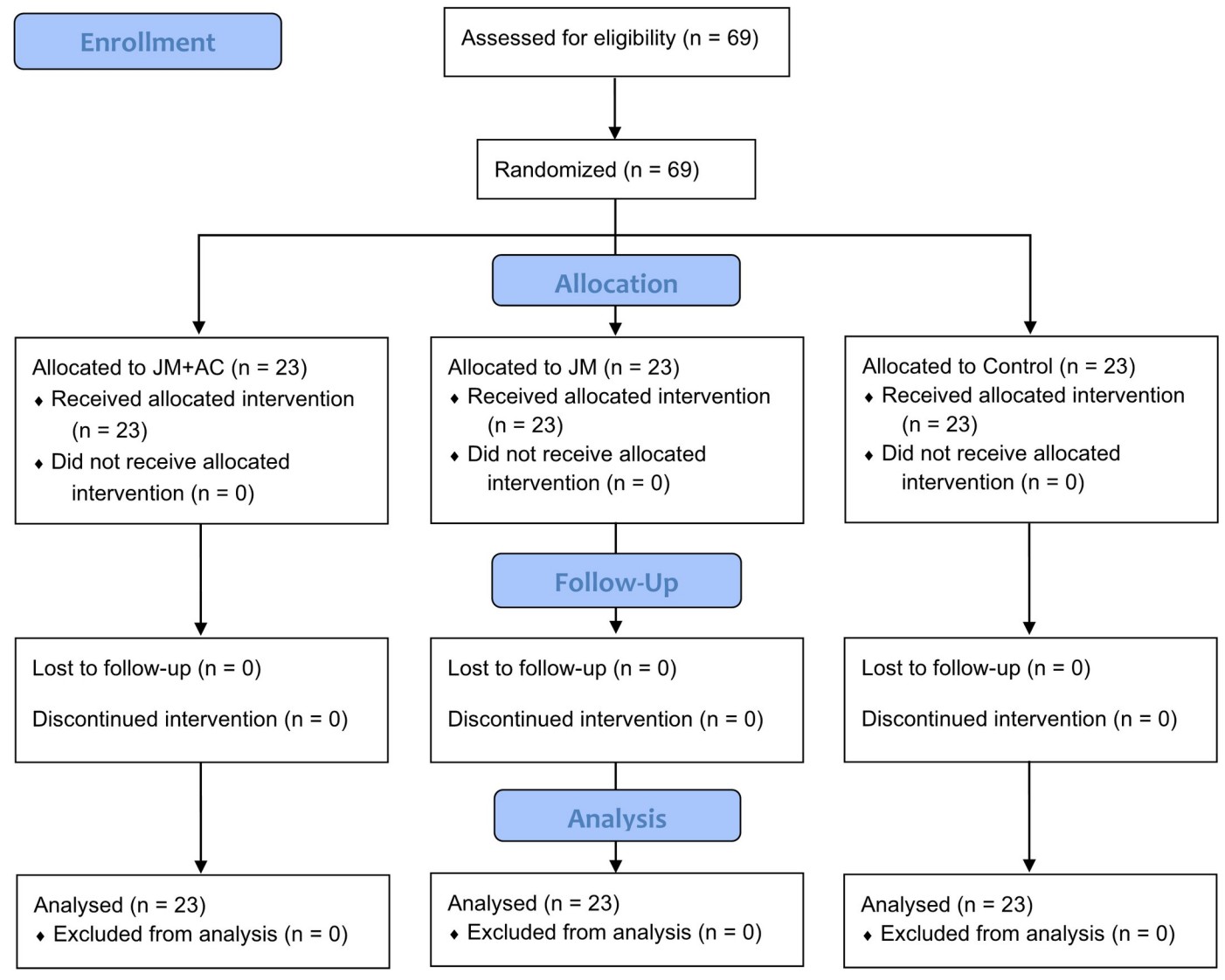

**Fig 1. CONSORT flow diagram.**

**Table 1. Participant characteristics.**

| Variables | JM+AC (n = 23) | JM (n = 23) | CON (n = 23) | p |
|---|---|---|---|---|
| Age (years) | 73.78 ± 6.52 | 74.13 ± 5.63 | 76.74 ± 6.62 | 0.23 |
| Sex (male/female) | 12 / 11 | 17 / 6 | 13 / 10 | 0.29 |
| Affected side (right/left) | 13 / 10 | 10 / 13 | 13 / 10 | 0.60 |
| Onset time (month) | 52.57 ± 24.39 | 61.35 ± 21.00 | 61.09 ± 22.20 | 0.33 |
| MMSE-K (score) | 26.17 ± 1.34 | 26.35 ± 1.44 | 25.87 ± 1.40 | 0.51 |

Values are means ± standard deviation.

MMSE-K: Korean mini mental state examination.

orthopedic disease or shoulder and knee joint surgery; and a length of ≥ 6.3 mm in the line bisection test; taking medications for diseases other than stroke, such as antiplatelet drugs and anticoagulants. The demographics and clinical characteristics of the study patients were summarized at inclusion (Table 1). No significant differences in the pretraining status of the control and experimental groups were noted.

## 3. Interventions

**3.1. Joint mobilization.** For 12 weeks, the JM+AC and JM groups received grades II and III JM for 30 min, twice weekly, to relieve pain in the shoulder and knee joints and to improve the ROM [12]. Shoulder JM was performed on the glenohumeral joint and scapulothoracic joint in the supine and prone positions. Lateral distraction, downward glide, posterior glide, and anterior glide were performed on the glenohumeral joint. Elevation, depression, protraction, retraction, upward rotation, and downward rotation were performed on the scapulothoracic joint. Knee JM was performed on the tibiofemoral and patellofemoral joints in the supine and prone positions. Lower distraction, posterior glide, and anterior glide were performed on the tibiofemoral joint. Downward, lateral, and medial glides were performed on the patellofemoral joint.

**3.2 Acupuncture.** The JM+AC group received acupuncture and moxibustion treatments around the shoulder and the knee joints for 30 min, once a week for 12 weeks. Acupuncture treatment was performed using disposable sterile stainless-steel needles (0.20 × 30 mm, Dongbang, Korea) for 15 min on average (Fig 2). The penetration depth varied according to the site and was 20 mm deep on average. For moxibustion treatment, mini moxibustion (Taegeuk, Korea) was performed for 15 min (Fig 2). The used acupoints include *binao* (LI14), *naohui* (TE13), *jianliao* (TE14), *tianliao* (TE15), *yanglingquan* (GB34), *yinlingquan* (SP9), *zusanli* (ST36), *fenglong* (ST40) (Fig 3). The positioning of acupoints is shown in Table 2 and is in accordance with the WHO Standard Acupuncture Locations.

## 4. Outcome measures

**4.1 Pain.** Pain severity was measured using a visual analog scale (VAS), the shoulder pain and disability index (SPADI), and the Korean Western Ontario McMaster Universities osteoarthritis index (KWOMAC) before and after 12 weeks of the experimental period. The VAS used a 100-mm line on which a pain-free state was 0 and severe pain was 100, and the reliability is 0.99 [22]. SPADI is an evaluation tool for assessing the degree of shoulder pain and disability that comprises 13 evaluation items scored on a scale of 0–10 [23]. A score above 100 (out of 130) indicates more shoulder pain and disability. The reliability is 0.94 [24]. KWOMAC, a Korean version of WOMAC [25], is an evaluation tool for assessing the degree of knee pain and function [26]. It consists of 24 evaluation items scored on a scale of 0–4 points and a total score of 96 points. Higher scores indicate a worse knee condition. The reliability is 0.96 [26].

**4.2 Physical function.** The range of motion (ROM), 10-meter walk test (10-MWT), functional gait assessment (FGA), manual function test (MFT) for measuring physical function, ADL, and instrumental ADL (IADL) were evaluated before and after 12 weeks of the experimental period.

A goniometer (Saehan Corporation, Korea) was used to measure the ROM of the shoulder and knee joint. Active ROM for shoulder flexion (SF), shoulder extension (SE), shoulder abduction (SAB), shoulder adduction (SAD), and knee flexion (KF) were measured. The measurement was performed three times, and the average value was used. The reliability is 0.89 [27].

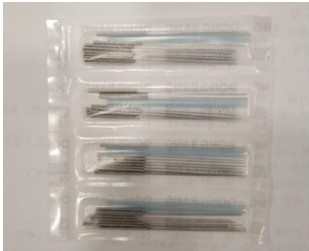 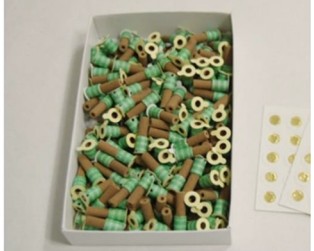

**Fig 2. Acupuncture and moxibustion treatment materials.** Disposable sterile needles (left picture). Disposable mini moxa rolls (right picture).

The 10-MWT requires the participant to walk independently over the middle 10 m portion on a 14-m path; the time is measured in seconds and the test is repeated three times. The reliability is 0.99 [28].

FGA was developed by Wrisley et al. [29] to evaluate gait and comprises 10 items scored on a scale of 0–3 points and a total score of 30 points. Specific items include walking on level surfaces, changing walking speed, turning the head to the side while walking, moving the head up and down while walking, walking with one foot as an axis, walking around obstacles, gait with a narrow base of support, walking with eyes closed, walking backward, and climbing stairs. The reliability is 0.93 [29].

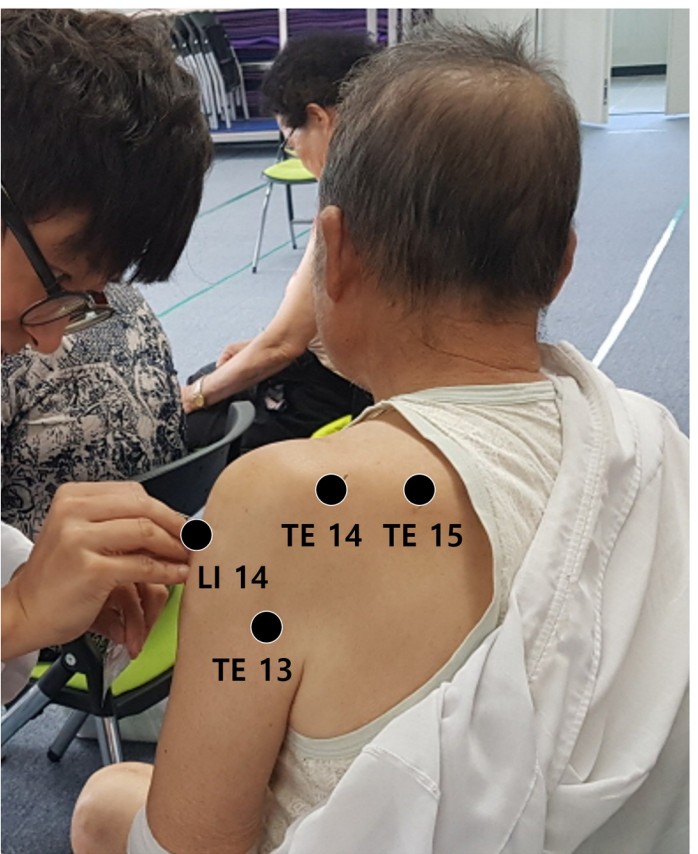 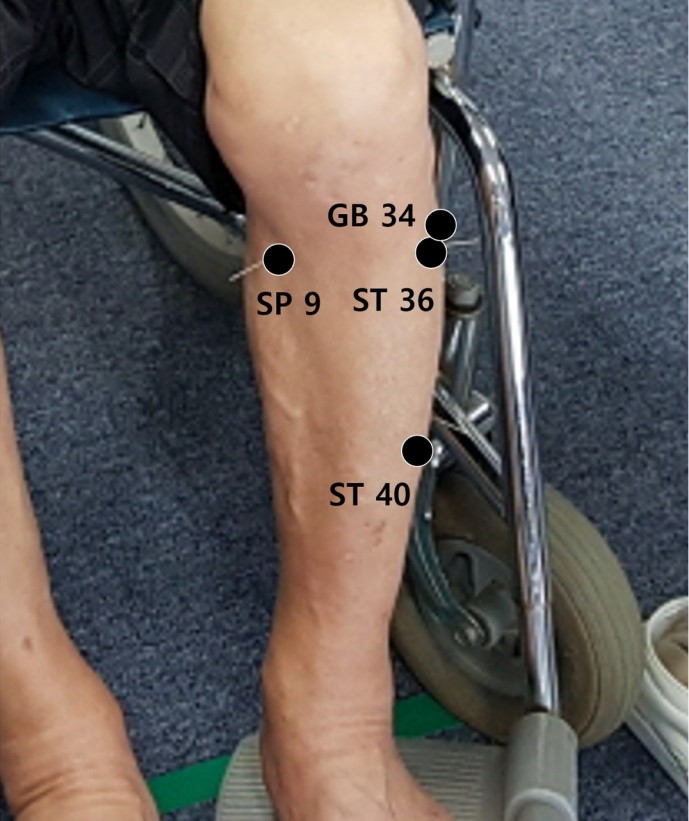

**Fig 3. Positioning of acupoints.** Shoulder acupoints (left picture). Knee acupoints (right picture).

**Table 2. Positioning of acupoints.**

| Acupoint | Location |
|---|---|
| *Binao* (LI14) | On the lateral aspect of the arm, just anterior to the border of the deltoid muscle, 7 B-cun superior to LI11 |
| *Naohui* (TE13) | On the posterior aspect of the arm, posteroinferior to the border of the deltoid muscle, 3 B-cun inferior to the acromial angle |
| *Jianliao* (TE14) | On the shoulder girdle, in the depression between the acromial angle and the greater tubercle of the humerus |
| *Tianliao* (TE15) | In the scapular region, in the depression superior to the superior angle of the scapula. |
| *Yanglingquan* (GB34) | On the fibular aspect of the leg, in the depression anterior and distal to the head of the fibula |
| *Yinlingquan* (SP9) | On the tibial aspect of the leg, in the depression between the inferior border of the medial condyle of the tibia and the medial border of the tibia |
| *Zusanli* (ST36) | On the anterior aspect of the leg, on the line connecting ST35 with ST41, 3 B-cun inferior to ST35 |
| *Fenglong* (ST40) | On the anterolateral aspect of the leg, lateral border of the tibialis anterior muscle, 8 B-cun superior to the prominence of the lateral malleolus |

MFT is a test tool developed to assess the overall condition of arm function in a stroke patient. It consists of eight items, and if subitem inspection is possible, it is measured as 1 point. The reliability is 0.95 [30].

ADL and IADL are tools developed by Won et al. [31] ADL comprising seven items covering dressing, washing the face, bathing, eating, moving, using the toilet, and controlling the bladder, with a total of 21 points. IADL comprises 30 items scored on a scale of 1–3. Ten items cover grooming, chores, preparing meals, washing clothes, going out a short distance, using transport, buying goods, managing money, using a phone, and taking medicine. Higher scores indicate lower dependence in evaluating the independence level in ADL. The reliability is 0.93 [31].

**4.3 Depression.** Depression was measured using the center for epidemiologic studies depression scale (CES-D) and the Beck depression inventory (BDI) before and after 12 weeks of the experimental period.

CES-D is a self-reported depression evaluation tool [32], comprising 20 evaluation items scored on a scale of 0–3 points. Scores higher than 21 (out of 60) indicate a higher frequency of depressive experiences. The reliability is 0.90 [32].

BDI is a tool for evaluating emotional, cognitive, motivational, and physiological depression, and the Korean version was used [33]. It comprises 21 evaluation items scored on a scale of 0–3 points and a total score of 63 points. The reliability is 0.84 [33].

## 5. Statistical analysis

The data were analyzed using SPSS 23.0 software (SPSS, Chicago, IL, USA). General characteristics prior to the interventions were analyzed using descriptive statistics. Two-way analysis of variance (2-way ANOVA) with repeated measures was performed to verify the difference between groups and time periods. Post hoc testing using Tukey's honestly significant difference (HSD) was performed to assess pairwise differences between groups. Additionally, a paired t-test was utilized to compare changes in pain, physical function, and depression before and after the experimental period within each group. Statistical significance was reached when $p < 0.05$.

## Results

### 1. Pain

The changes in pain severity are shown in Table 3. SPADI and KWOMAC both showed statistically significant group interaction effects (F(2,66) = 25.098 and 17.965, respectively, p < 0.05). The VAS was not statistically significant between groups, but SPADI decreased significantly in the JM+AC group compared to the JM group and the control group (p < 0.05), and KWOMAC decreased significantly in the JM+AC group compared to the control group (p < 0.05). The VAS, SPADI, and KWOMAC were decreased significantly in the JM+AC and JM groups after 12 weeks of intervention (p < 0.05).

### 2. Physical function

The changes in physical functions are shown in Table 4.

**2.1 Range of motion.** SF, SE, SAB, and SAD were not statistically significant among groups, but KF increased significantly in the JM+AC and JM groups compared to the control group (p < 0.05). Within the JM+AC group, SF, SE, SAB, and SAD increased significantly (p < 0.05), whereas KF increased significantly in the JM+AC and JM groups after 12 weeks of intervention (p < 0.05).

**2.2 Gait.** The 10-MWT and FGA were not statistically significant between groups, but 10-MWT and FGA improved significantly in the JM+AC and JM groups after 12 weeks of intervention (p < 0.05).

**2.3 Activities of daily living.** MFT, ADL and IADL both showed statistically significant group interaction effects (F(2,66) = 21.421, 16.549, and 8.513, respectively, p < 0.05). MFT increased significantly in the JM+AC group compared to the JM group and control groups (p < 0.05), as well as in the JM group compared to the control group (p < 0.05). ADL increased significantly in the JM+AC and JM groups compared to the control group (p < 0.05), and IADL increased significantly in the JM+AC group compared to the JM group and the control group (p < 0.05). Within the JM+AC and JM groups, MFT and ADL increased significantly (p < 0.05), whereas IADL increased significantly in the JM+AC group after 12 weeks of intervention (p < 0.05).

### 3. Depression

The changes in depression are shown in Table 5. CES-D and BDI both showed statistically significant group interaction effects (F(2,66) = 7.323 and 18.495, respectively, p < 0.05). CES-D and BDI decreased significantly in the JM+AC and JM groups compared to the control group (p < 0.05), and CES-D and BDI decreased significantly in the JM+AC group after 12 weeks of intervention (p < 0.05).

## Discussion

The main finding of this study was that JM combined with AC treatment improved the subjective pain, depression, and physical function of poststroke patients more effectively than JM alone.

Shoulder and knee pain impedes rehabilitation and interferes with daily activities in poststroke patients [6–8]. Kaltenborn et al. [12] indicated that JM was effective in relieves shoulder and knee pain through direct movement of the joints. In AC treatments, acupuncture treatment is reported to increase pain threshold by stimulating the central nervous system, and moxibustion treatment has an analgesic effect through thermal stimulation [18, 34]. Additionally, previous studies reported that AC treatment reduced SPADI and KOMAC [35, 36].

**Table 3. Pain before and after joint mobilization and joint mobilization combined with acupuncture.**

| Variables | JM+AC (n = 23) | | | JM (n = 23) | | | CON (n = 23) | | | F$^b$ (p) | ES |
|---|---|---|---|---|---|---|---|---|---|---|---|
| Period | pre | post | T$^a$ (p) | pre | post | T$^a$ (p) | pre | post | T$^a$ (p) | | |
| VAS (mm) | 75.65 ± 18.55 | 56.13 ± 5.80 | 5.18* (0.000) | 63.48 ± 3.45 | 56.30 ± 4.35 | 2.32* (0.030) | 62.83 ± 14.99 | 67.17 ± 18.52 | −1.50 (0.148) | G = 0.965 (0.386) T = 15.509* (0.000) G×T = 13.275 (0.000) | G = 0.944 T = 0.190 G×T = 0.287 |
| SPADI (score) | 68.90 ± 0.12 | 63.71 ± 8.40†‡ | 6.11* (0.000) | 58.16 ± 11.92 | 56.79 ± 11.98 | 3.32* (0.003) | 58.56 ± 10.32 | 58.86 ± 10.27 | −1.31 (0.205) | G = 4.783* (0.011) T = 41.406* (0.000) G×T = 25.098* (0.000) | G = 0.127 T = 0.386 G×T = 0.432 |
| KWOMAC (score) | 73.40 ± 5.22 | 51.88 ± 12.46† | 5.93* (0.000) | 68.65 ± 11.95 | 59.57 ± 9.80 | 3.07* (0.006) | 68.61 ± 11.40 | 71.43 ± 12.87 | −1.70 (0.104) | G = 0.042* (0.042) T = 31.183* (0.000) G×T = 17.965* (0.000) | G = 0.091 T = 0.321 G×T = 0.352 |

Values are means ± standard deviation.

VAS: visual analog scale, SPADI: shoulder pain and disability index, KWOMAC: Korean Western Ontario and McMaster Universities osteoarthritis index.

$^a$T: comparison within groups

$^b$F: comparison between groups.

*p < 0.05 vs. pre-exercise.

†p < 0.05 vs. CON.

‡p < 0.05 vs. JM.

Taken together, it was confirmed that JM combined with AC treatment had a more positive effect on pain in poststroke patients than JM alone.

Kluding and Santos [37] stated that the JM technique improved movement without changing in joint ROM. Alternatively, Choi et al. [38] reported that AC treatment reduces pain and increases joint flexibility by regenerating the surrounding soft tissue. We found that there was a significant difference in ROM before and after the intervention in the JM+ AC group, but not in the JM group in this study. These results may imply that it is difficult to improve joint ROM with JM alone.

In order to be independent and participate in society, gait is crucial, as was mentioned in the introduction. JM showed a positive effect on gait by improving active ROM which is the ROM that can be achieved by patient himself/herself [39], and AC treatment stimulated muscles and nerves to improve gait [40]. Additionally, previous studies stated that acupuncture had a beneficial impact on the energy production of joints during walking [41]. In this study, gait was significantly improved after the intervention in the JM+ AC and JM groups, but there was no difference between groups. Generally, gait necessitates movement of hip, knee, and ankle joints. The lower extremity interventions in this study were limited to the knee joint, which may not have been sufficient to influence group effect comparisons. In the future, it would be prudent to intervene in all joints of lower extremity.

We employed several measurements to access physical function, which is important for daily living. MFT is an assessment of overall arm function [30], SPADI is for shoulder pain and disability [23]. IADL is for evaluation of dressing, toilet using, and movement [31]. It is thought that the reduction of SPADI restored arm function in poststroke patients and affected

**Table 4. Physical function before and after joint mobilization and joint mobilization combined with acupuncture.**

| Variables | | JM+AC (n = 23) | | | JM (n = 23) | | | CON (n = 23) | | | $F^b$ (p) | ES |
|---|---|---|---|---|---|---|---|---|---|---|---|---|
| Period | | pre | post | $T^a$ (p) | pre | post | $T^a$ (p) | pre | post | $T^a$ (p) | | |
| ROM | SF (˚) | 119.91 ± 14.00 | 125.26 ± 13.26 | −4.74* (0.000) | 128.17 ± 14.46 | 129.17 ± 13.21 | −1.17 (0.253) | 128.65 ± 3.02 | 124.48 ± 3.30 | 4.18* (0.000) | G = 1.117 (0.333) T = 1.577 (0.214) G×T = 22.745 (0.000) | G = 0.033 T = 0.023 G×T = 0.408 |
| | SE (˚) | 33.70 ± 8.13 | 38.96 ± 6.64 | −5.79* (0.000) | 37.52 ± 8.63 | 37.83 ± 1.40 | −0.36 (0.724) | 34.48 ± 1.76 | 32.39 ± 1.40 | 2.36* (0.028) | G = 2.029 (0.140) T = 5.174* (0.026) G×T = 18.022 (0.000) | G = 0.058 T = 0.073 G×T = 0.353 |
| | SAB (˚) | 109.13 ± 11.93 | 114.70 ± 10.17 | −6.26* (0.000) | 117.44 ± 2.46 | 118.52 ± 2.02 | −1.38 (0.183) | 112.21 ± 2.41 | 110.22 ± 2.64 | 2.26* (0.034) | G = 2.558 (0.085) T = 9.853* (0.003) G×T = 19.762 (0.000) | G = 0.072 T = 0.130 G×T = 0.375 |
| | SAD (˚) | 40.83 ± 6.34 | 47.65 ± 6.77 | −7.13* (0.000) | 39.91 ± 1.64 | 42.39 ± 1.47 | −1.89 (0.072) | 41.96 ± 1.10 | 40.00 ± 0.89 | 2.20* (0.038) | G = 2.274 (0.111) T = 15.791* (0.000) G×T = 16.921 (0.000) | G = 0.064 T = 0.193 G×T = 0.339 |
| | KF (˚) | 117.31 ± 11.68 | 127.17 ± 8.71† | −5.49* (0.000) | 119.91 ± 2.02 | 125.65 ± 1.70† | −4.37* (0.000) | 112.83 ± 2.61 | 112.39 ± 2.65 | 0.44 (0.665) | G = 7.266* (0.001) T = 38.774* (0.000) G×T = 13.587* (0.000) | G = 0.180 T = 0.370 G×T = 0.292 |
| GAIT | 10-MWT (s) | 25.83 ± 11.49 | 14.81 ± 8.45 | 7.44* (0.000) | 23.23 ± 6.23 | 18.13 ± 6.49 | 4.97* (0.000) | 24.55 ± 9.58 | 24.04 ± 9.92 | 0.98 (0.336) | G = 1.539 (0.222) T = 78.657* (0.000) G×T = 23.696 (0.000) | G = 0.045 T = 0.544 G×T = 0.418 |
| | FGA (score) | 18.78 ± 3.28 | 23.13 ± 2.70 | −5.90* (0.000) | 19.04 ± 3.04 | 22.00 ± 2.70 | −4.22* (0.000) | 20.04 ± 3.83 | 19.74 ± 4.17 | 0.79 (0.437) | G = 0.747 (0.478) T = 41.389* (0.000) G×T = 14.449 (0.000) | G = 0.022 T = 0.385 G×T = 0.305 |
| ADL | MFT (score) | 20.78 ± 1.57 | 22.91 ± 1.73†‡ | −6.16* (0.000) | 20.43 ± 1.41 | 21.52 ± 1.62† | −4.64* (0.000) | 20.00 ± 1.71 | 19.52 ± 1.68 | 1.85 (0.077) | G = 11.674* (0.000) T = 31.074* (0.000) G×T = 21.421* (0.000) | G = 0.261 T = 0.320 G×T = 0.394 |
| | ADL (score) | 17.57 ± 0.52 | 20.22 ± 1.88† | −5.42* (0.000) | 17.13 ± 0.39 | 19.30 ± 0.19† | −7.61* (0.000) | 16.65 ± 0.36 | 16.57 ± 0.40 | 0.33 (0.740) | G = 13.931* (0.000) T = 57.879* (0.000) G×T = 16.549* (0.000) | G = 0.297 T = 0.467 G×T = 0.334 |
| | IADL (score) | 24.35 ± 0.90 | 26.96 ± 0.39†‡ | −1.13* (0.001) | 23.13 ± 0.81 | 23.57 ± 0.72 | -0.93 (0.360) | 22.96 ± 0.80 | 22.70 ± 0.70 | 1.03 (0.314) | G = 4.773* (0.012) T = 9.805* (0.003) G×T = 8.513* (0.001) | G = 0.126 T = 0.129 G×T = 0.205 |

Values are means ± standard deviation.

ROM: range of motion, SF: shoulder flexion, SE: shoulder extension, SAB: shoulder abduction, SAD: shoulder adduction, KF: knee flexion, KE: knee extension, 10-MWT: 10-meter walking speed test, FGA: functional gait assessment, MFT: manual function test, ADL: activities of daily living scale, IADL: instrumental activities of daily living scale.

[a] T: comparison within groups

[b] F: comparison between groups.

*p < 0.05 vs. pre-exercise.

†p < 0.05 vs. CON.

‡p < 0.05 vs. JM.

**Table 5. Depression before and after joint mobilization and joint mobilization combined with acupuncture.**

| Variables | JM+AC (n = 23) | | | JM (n = 23) | | | CON (n = 23) | | | F[b] (p) | ES |
|---|---|---|---|---|---|---|---|---|---|---|---|
| Period | pre | post | T[a] (p) | pre | post | T[a] (p) | pre | post | T[a] (p) | | |
| CES-D (score) | 26.83 ± 8.63 | 20.30 ± 8.87† | 5.24* (0.000) | 20.48 ± 9.57 | 20.43 ± 8.88 | 0.06 (0.952) | 27.91 ± 8.96 | 33.00 ± 8.77 | −3.60* (0.002) | G = 7.106* (0.002) T = 0.007 (0.935) G×T = 7.323* (0.001) | G = 0.177 T = 0.000 G×T = 0.182 |
| BDI (score) | 32.48 ± 11.04 | 22.35 ± 7.97† | 5.49* (0.000) | 30.61 ± 12.23 | 29.96 ± 10.99 | 1.19 (0.249) | 33.70 ± 9.49 | 39.43 ± 9.15 | −3.33* (0.003) | G = 5.971* (0.004) T = 9.863* (0.003) G×T = 18.495* (0.000) | G = 0.153 T = 0.130 G×T = 0.359 |

Values are means ± standard deviation.

CES-D: center for epidemiologic studies depression scale, BDI: Beck depression inventory.

[a]T: comparison within groups

[b]F: comparison between groups.

*p < 0.05 vs. pre-exercise.

†p < 0.05 vs. CON.

‡p < 0.05 vs. JM.

MFT and IADL. Previous research has shown that acupuncture has little effect on improving the lifestyle of stroke patients [42], but it is believed that it had a positive effect on IADL in this study because joint mobilization was combined with acupuncture. Arm function and daily life activities of the patients in the JM+ AC group might be improved through the increase in muscle activity induced by JM and the recovery of neurological damage by AC treatment.

Depression is associated with pain in poststroke patients [6], and depression in these patients are associated with mortality, suicide risk, and quality of life [43–45]. Thus, depression must be verified in rehabilitation process for functional recovery after stroke, in general. The CES-D and BDI scores in the JM+ AC group were significantly decreased after 12 weeks of the intervention, but not in the JM group. This is most likely due to AC treatment to alleviate pain and improve ADL. A previous study showed that JM improved depression by reducing stress factors through improvement of unstable posture [46]. Furthermore, it has been reported that acupuncture treatment was effective in suppressing changes in the autonomic nervous system caused by mental stress [47]. It has also been reported that moxibustion treatment relaxed and stabilized the body through its thermal action [18]. Thus, it is considered that acupuncture and moxibustion treatment of AC reduced depression through autonomic nerve inhibition and thermal stimulation.

## Limitation

There are several limitations in our study. Firstly, the majority part of subjects in this study were older farmers living in rural areas, and it is not uncertain whether the results of this study can be applied to other elderly population living in urban areas. Secondly, it might be better if we assessed the effect of the treatment more frequently because the results of this study were obtained before and after the 12 weeks of intervention period.

Furthermore, we acknowledge that there are some constraints and limitations that should be taken into account when interpreting the results. First off, the study's relatively small sample size may limit how broadly the findings can be applied to larger populations. Secondly, caution is advised due to the potential influence of uncontrolled confounding factors, even though the observed differences in age and length of time since onset between the JM+AC group and other groups were statistically significant.

## Conclusion

Improvements in pain, depression, and physical function of poststroke patients in the JM+ AC group were significantly greater than those in the JM and the control groups, suggesting JM combined with AC was more effective in improving not only pain but also the ability to physical function. JM combined with AC may also be helpful for poststroke patients who have depression and decreased physical function due to chronic neuropathic pain. Future studies should reveal the physiological assessment of muscle strength and the effect of JM combined with AC on muscle tone based on changes in physical function in stroke patients with chronic neuropathic pain.

## Supporting information

**S1 Checklist. CONSORT checklist.** Checklist of information to include when reporting a randomized trial.
(DOC)

**S1 File. Trial protocol in Korean.**
(PDF)

**S2 File. Trial protocol in English.**
(PDF)

## Acknowledgments

We appreciate Yeon-Oh Kim and Jong-Bin Choi of the Dangjin City Public Health Center for their assistance with this experiment.

## Author Contributions

**Conceptualization:** Ji-Eun Lee.

**Data curation:** Ji-Eun Lee.

**Methodology:** Ji-Eun Lee, Takayuki Akimoto, Ho-Seong Lee.

**Resources:** Ji-Eun Lee, Jisuk Chang.

**Writing – original draft:** Ji-Eun Lee, Takayuki Akimoto.

**Writing – review & editing:** Takayuki Akimoto, Ho-Seong Lee.

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
