## [Decision Letter · Decision Letter 0]

29 May 2023

PONE-D-23-03014Effects of joint mobilization combined with acupuncture on pain, physical function, and depression in stroke patients with chronic neuropathic pain: a randomized controlled trialPLOS ONE

Dear Dr. Lee,

Thank you for submitting your manuscript to PLOS ONE. After careful consideration, we feel that it has merit but does not fully meet PLOS ONE’s publication criteria as it currently stands. Therefore, we invite you to submit a revised version of the manuscript that addresses the points raised during the review process.

ACADEMIC EDITOR:As a scientific study of acupuncture and moxibustion, the most fatal problem in this paper is, as pointed out by reviewer 2, that acupuncture and moxibustion MUST provide consistent basic acupoints, and show and write the name and location of the acupoints, e.g. based on "WHO Standard Acupuncture Point Locations ( https://apps.who.int/iris/handle/10665/353407)". Also, if possible, the authors could provide specific images and videos of joint mobilization techniques.

We look forward to receiving your revised manuscript.

Kind regards,

Tadashi Kobayashi, M.D., Ph.D

Academic Editor

PLOS ONE

2. We note that the original protocol that you have uploaded as a Supporting Information file contains an institutional logo. As this logo is likely copyrighted, we ask that you please remove it from this file and upload an updated version upon resubmission.

Additional Editor Comments:

As a scientific study of acupuncture and moxibustion, the most fatal problem in this paper is, as pointed out by reviewer 2, that acupuncture and moxibustion MUST provide consistent basic acupoints, and show and write the name and location of the acupoints, e.g.　based on "WHO Standard Acupuncture Point Locations ( https://apps.who.int/iris/handle/10665/353407)". Also, if possible, the authors could provide specific images and videos of joint mobilization techniques.

Reviewers' comments:

Reviewer's Responses to Questions

**Comments to the Author**

1. Is the manuscript technically sound, and do the data support the conclusions?

Reviewer #1: Yes

Reviewer #2: No

Reviewer #3: Yes

2. Has the statistical analysis been performed appropriately and rigorously? 

Reviewer #1: Yes

Reviewer #2: No

Reviewer #3: Yes

3. Have the authors made all data underlying the findings in their manuscript fully available?

Reviewer #1: Yes

Reviewer #2: Yes

Reviewer #3: Yes

4. Is the manuscript presented in an intelligible fashion and written in standard English?

Reviewer #1: Yes

Reviewer #2: Yes

Reviewer #3: Yes

5. Review Comments to the Author

Reviewer #1: This manuscript reports the results of a study in post-stroke patients of two competing treatments and a control treatment. The manuscript is generally well-written and the study appears to be well-performed.

Line 49: Change to "post-stroke".

Line 210: Delete "statistically".

Please provide more detail on the methodology used to perform the two-way analysis with repeated measures. Was this implemented within the linear mixed model framework? If so please describe the model in terms of fixed effects, G-side terms, and R-side terms (if any). Or was this implemented within the older framework using Greenhouse-Geisser or Huynh-Feldt correction? If so please describe the methodology used to assess sphericity. Or was this implemented using the older framework using a multivariate analysis of variance? If so, please describe the tests used.

Line 228: For this and other tables, please provide the analysis of variance p-values for the terms in the model --- or at least consider providing these as supplementary information. Please footnote the table to provide the statistical tests or multiple comparison methods used.

Note that pairwise comparison methods that control the experimentwise error rate may be used regardless of the analysis of variance results. However, it is best to assess the data for interaction effects before assuming the main effects structure.

Line 235: Change "between" to "among". Or clarify that this "between" refers to pairwise comparisons.

Line 314: Change to "The majority... were older farmers...".

Reviewer #2: 1. In my opinion, a reasonable randomized controlled trial should provide a formula for calculating sample size, but this paper does not2. Personally, I think acupuncture and moxibustion should provide consistent basic acupoints. This paper does not clearly give the name and location of the acupoints taken

Reviewer #3: Abstract: To investigate....(AC) for THE treatment OF pain, ... (missing words)

Introduction line 44 "Stroke is one of (how many? what percentage)?

This sentence is confusing (line 73): "The convex-concave rule states that gliding on a convex joint surface should be done the opposite way the bone moves, and gliding on a concave joint surface should be done the opposite way." Not sure why both state "opposite way."

Reference for line 80?

Table 1: the JM+AC group were younger, with less time since onset. Although not a significant p value, these differences need to be addressed.

Line 150: Was there standard acupuncture points used? Or was the AC treatment individualized?

6. PLOS authors have the option to publish the peer review history of their article (what does this mean?). If published, this will include your full peer review and any attached files.

Reviewer #1: No

Reviewer #2: No

Reviewer #3: **Yes: **Jennifer E Brett

---

## [Author Response · Author response to Decision Letter 0]

16 Jun 2023

Response to Reviewers

PONE-D-23-03014

Effects of joint mobilization combined with acupuncture on pain, physical function, and depression in stroke patients with chronic neuropathic pain: a randomized controlled trial

PLOS ONE

Editor:

Thank you for submitting your manuscript to PLOS ONE. After careful consideration, we feel that it has merit but does not fully meet PLOS ONE’s publication criteria as it currently stands. Therefore, we invite you to submit a revised version of the manuscript that addresses the points raised during the review process.

As a scientific study of acupuncture and moxibustion, the most fatal problem in this paper is, as pointed out by reviewer 2, that acupuncture and moxibustion MUST provide consistent basic acupoints, and show and write the name and location of the acupoints, e.g. based on "WHO Standard Acupuncture Point Locations ( https://apps.who.int/iris/handle/10665/353407)". Also, if possible, the authors could provide specific images and videos of joint mobilization techniques.

Response: Thank you very much for allowing us to revise our manuscript. We have carefully considered the comments and tried our best to address every one of them. We hope the changes and responses meet the expectation of PLOS ONE and the manuscript is now suitable for publication.

Reviewer #1:

This manuscript reports the results of a study in post-stroke patients of two competing treatments and a control treatment. The manuscript is generally well-written and the study appears to be well-performed.

Response: We appreciate your precious time in reviewing our paper and providing valuable comments. 

Line 49: Change to "post-stroke".

Response: We have changed on Page 3, Line 49. 

Line 210: Delete "statistically".

Response: We have deleted it.

Please provide more detail on the methodology used to perform the two-way analysis with repeated measures. 

Response: We appreciate your thoughtful comment about our analysis. We have added some sentences regarding statistics on Page 13, Line 237-241.

Was this implemented within the linear mixed model framework? If so please describe the model in terms of fixed effects, G-side terms, and R-side terms (if any). Or was this implemented within the older framework using Greenhouse-Geisser or Huynh-Feldt correction? If so please describe the methodology used to assess sphericity. Or was this implemented using the older framework using a multivariate analysis of variance? If so, please describe the tests used.

Response: We took advantage of 2-way ANOVA with repeated measures instead of using a linear mixed model framework or specific sphericity correction methods in our analysis. We have added some sentences regarding this on Page 13, Line 237-241.

Line 228: For this and other tables, please provide the analysis of variance p-values for the terms in the model --- or at least consider providing these as supplementary information. Please footnote the table to provide the statistical tests or multiple comparison methods used.

Response: We have added these, accordingly.

Note that pairwise comparison methods that control the experimentwise error rate may be used regardless of the analysis of variance results. However, it is best to assess the data for interaction effects before assuming the main effects structure.

Response: Thank you for the valuable comment. We added the results for interaction effects to the manuscript on pages 14-19.

Line 235: Change "between" to "among". Or clarify that this "between" refers to pairwise comparisons.

Response: We have corrected it.

Line 314: Change to "The majority... were older farmers...".

Response: We have corrected it. Thank you.

Reviewer #2: 

1. In my opinion, a reasonable randomized controlled trial should provide a formula for calculating sample size, but this paper does not.

Response: Thank you very much for the comment. We have added it on Page 6, Line 109-115.

2. Personally, I think acupuncture and moxibustion should provide consistent basic acupoints. This paper does not clearly give the name and location of the acupoints taken.

Response: Thank you very much for the valuable comment. We added the information on Page 9, Line 168-172 and revised the manuscript accordingly.

Reviewer #3: 

Abstract: To investigate....(AC) for THE treatment OF pain, ... (missing words)

Response: Thank you very much for the comment. We have corrected it.

Introduction line 44 "Stroke is one of (how many? what percentage)?

Response: Thank you very much for the comment. We have changed it.

This sentence is confusing (line 73): "The convex-concave rule states that gliding on a convex joint surface should be done the opposite way the bone moves, and gliding on a concave joint surface should be done the opposite way." Not sure why both state "opposite way."

Response: We are sorry for the confusion. We have corrected the sentence. Thank you.

Reference for line 80?

Response: We have corrected the reference. Thank you.

Table 1: the JM+AC group were younger, with less time since onset. Although not a significant p value, these differences need to be addressed.

Response: We appreciate your comment. We are aware of the limitations in interpreting the differences in age and time since onset found in the JM+AC group. We are afraid that it may be difficult to generalize the results to other populations due to the small sample size in our study. We accordingly made changes to the limitations of the manuscript on Page 23, Line 351-357.

Line 150: Was there standard acupuncture points used? Or was the AC treatment individualized?

Response: Thank you very much for the comment. We added the information on Page 9, Line 168-172 and revised the manuscript accordingly.

---

## [Decision Letter · Decision Letter 1]

12 Jul 2023

Effects of joint mobilization combined with acupuncture on pain, physical function, and depression in stroke patients with chronic neuropathic pain: a randomized controlled trial

PONE-D-23-03014R1

Dear Dr. Lee,

We’re pleased to inform you that your manuscript has been judged scientifically suitable for publication and will be formally accepted for publication once it meets all outstanding technical requirements.

Kind regards,

Tadashi Kobayashi, M.D., Ph.D

Academic Editor

PLOS ONE

Additional Editor Comments (optional):

The authors have amended adequately.

Reviewers' comments:

Reviewer's Responses to Questions

**Comments to the Author**

1. If the authors have adequately addressed your comments raised in a previous round of review and you feel that this manuscript is now acceptable for publication, you may indicate that here to bypass the “Comments to the Author” section, enter your conflict of interest statement in the “Confidential to Editor” section, and submit your "Accept" recommendation.

Reviewer #1: All comments have been addressed

Reviewer #2: All comments have been addressed

Reviewer #3: All comments have been addressed

2. Is the manuscript technically sound, and do the data support the conclusions?

Reviewer #1: (No Response)

Reviewer #2: (No Response)

Reviewer #3: Yes

3. Has the statistical analysis been performed appropriately and rigorously? 

Reviewer #1: (No Response)

Reviewer #2: (No Response)

Reviewer #3: Yes

4. Have the authors made all data underlying the findings in their manuscript fully available?

Reviewer #1: (No Response)

Reviewer #2: (No Response)

Reviewer #3: Yes

5. Is the manuscript presented in an intelligible fashion and written in standard English?

Reviewer #1: (No Response)

Reviewer #2: (No Response)

Reviewer #3: Yes

6. Review Comments to the Author

Reviewer #1: (No Response)

Reviewer #2: (No Response)

Reviewer #3: (No Response)

7. PLOS authors have the option to publish the peer review history of their article (what does this mean?). If published, this will include your full peer review and any attached files.

Reviewer #1: No

Reviewer #2: No

Reviewer #3: No

---

## [Editor Report · Acceptance letter]

15 Aug 2023

PONE-D-23-03014R1 

Effects of joint mobilization combined with acupuncture on pain, physical function, and depression in stroke patients with chronic neuropathic pain: a randomized controlled trial 

Dear Dr. Lee:

I'm pleased to inform you that your manuscript has been deemed suitable for publication in PLOS ONE. Congratulations! Your manuscript is now with our production department. 

Kind regards, 

on behalf of

Dr. Tadashi Kobayashi 

Academic Editor

PLOS ONE